# The Enzymatic Hydrolysis of Human Milk Oligosaccharides and Prebiotic Sugars from LAB Isolated from Breast Milk

**DOI:** 10.3390/microorganisms11081904

**Published:** 2023-07-27

**Authors:** Daniela Mollova, Tonka Vasileva, Veselin Bivolarski, Ilia Iliev

**Affiliations:** Department of Biochemistry and Microbiology, Faculty of Biology, Plovdiv University “Paisii Hilendarski”, 4000 Plovdiv, Bulgaria; vasileva@uni-plovdiv.bg (T.V.); bivolarski@uni-plovdiv.bg (V.B.); iliailiev@uni-plovdiv.bg (I.I.)

**Keywords:** human milk oligosaccharides, *Lactobacillus*, prebiotics, enzymes, breast milk

## Abstract

Breastfeeding is essential in the first months of a newborn’s life. Breast milk is a source of crucial macronutrients, prebiotic oligosaccharides, and potential probiotic strains of bacteria. Oligosaccharides from breast milk (HMOs) are a significant part of the composition of breast milk and represent a complex of digestible sugars. This study aims to elucidate the enzymatic hydrolysis of these oligosaccharides and other prebiotics by the bacteria present in breast milk. We used modified methods to isolate oligosaccharides (HMOs) from human milk. Using unique techniques, we isolated and identified different bacteria from breast milk, mainly *Lactobacillus fermentum*. Using enzymatic analyses, we established the participation of α-fucosidase, α-glucosidase, β-galactosidase, and β-glucosidase from breast milk bacteria in the hydrolysis of prebiotic sugars. We also optimized the scheme for isolating oligosaccharides from breast milk by putting the lyophilized product into different food media. We found that the oligosaccharides from breast milk (HMOs) are a potent inducer for the secretion of the studied bacterial enzymes. Also, we found that all the lactobacilli strains we studied in detail could digest mucin-linked glycans. The degradation of these sugars is perhaps a built-in defense mechanism in cases where other sugars are lacking in the environment. We also determined fucosidase activity in some of the isolated strains. We recorded the highest values (2.5 U/mg in *L. fermentum* ss8) when the medium’s oligosaccharides isolated from breast milk were present. *Lactobacilli* and *Bifidobacteria* supplied with breast milk are the first colonizers in most cases in the gastrointestinal tract of the newborn. The presence and study of different genes for synthesizing other enzyme systems and transporters of various sugars in this type of bacteria are essential.

## 1. Introduction

Newborn infants thrive best when fed with human breast milk, which is widely acknowledged as the optimal feeding method. It is renowned for its numerous advantages in promoting infant health, growth, and development. Both the World Health Organization (WHO) and The United Nations Children’s Fund (UNICEF) advocate for the initiation of breastfeeding within the first hour after birth and recommend exclusive breastfeeding for the initial six months of a baby’s life [1].

Breast milk contains the perfect balance of essential macro and micronutrients, offering comprehensive nourishment for infants. Apart from these vital nutrients, breast milk also contains various bioactive and immune components like antibodies, lysozyme, growth factors, antimicrobial peptides, microRNAs, stem cells, and human milk oligosaccharides (HMOs). These elements are likely to play a crucial role in shaping the developing infant’s immune system and providing protection against harmful pathogens [2,3,4]. 

Moreover, breast milk contains advantageous microorganisms like *Bifidobacterium* and *Lactobacillus*, renowned for their health benefits. Human milk oligosaccharides (HMOs) play a role in promoting the growth of these potentially beneficial probiotic bacteria in the infant’s digestive system [5,6,7]. Therefore, consuming breast milk during lactation helps in establishing the infant’s gut microbiota, leading to the enhancement and maturation of the newborn’s immune system [8].

In recent research, it has been discovered that both colostrum and breast milk serve as ongoing suppliers of commensal, mutualistic, and possibly probiotic bacteria within the baby’s gastrointestinal tract [9,10,11,12,13,14,15,16]. This discovery holds significance as it challenges the traditional belief that intramammary human milk is devoid of bacteria. Human milk is a major contributor to the bacterial population in the gut of breastfed infants, as a baby consuming around 800 mL of milk per day would be ingesting between 1 × 10^5^ and 1 × 10^7^ bacteria on a daily basis [10].

Emphasizing the significance of human milk oligosaccharides (HMOs), it is important to note that they play a vital role in promoting the diversity of the microbiota in the infant’s gastrointestinal system [17]. HMOs, ranking as the third most abundant solid element in milk, possess a complex structure that makes them indigestible for the host. However, the hydrolytic enzymes present in the colonic microbiota of infants can break them down [18]. Earlier studies have shown that certain strains of bifidobacteria and lactobacilli utilize particular human milk oligosaccharides (HMOs) that are produced during the early stages of lactation [19]. Moreover, recent investigations into the genome sequences of two bifidobacterial strains have uncovered numerous adaptations that enable them to efficiently make use of milk constituents in the infant’s microbiome [20,21].

### 1.1. Functions of Human Milk Bacteria in the Infant’s Gut

Some recent studies have shown that human milk bacteria may play several roles in the infant’s gut. First of all, they can contribute to a reduction in the incidence and severity of infections in the breastfed infant by different mechanisms, such as competitive exclusion, production of antimicrobial compounds [15,22], or improvement of the intestinal barrier function by increasing mucin production and reducing intestinal permeability [5]. Breast milk bacteria may also participate in the correct maturation of the infant immune system since some strains can modulate both natural and acquired immune responses in mice and humans [23,24,25]. According to recent research, it was verified that *L. fermentum* CECT 5716 and *L. salivarius* CECT 5713 exhibited diverse effects on the immune system. The study found that these two strains acted as strong stimulators of NK cells and moderately activated CD4+ and CD8+ T cells, as well as regulatory T cells. As a result, they significantly influenced both the innate and acquired immune responses. Additionally, these strains strongly induced a broad spectrum of pro- and anti-inflammatory cytokines and chemokines.

Ultimately, the bacteria found in human milk possess significant potential in influencing the infant’s metabolism. Certain lactobacilli and Bifidobacteria, which are present in human milk, play a role in establishing a beneficial and unique microbial community in the infant’s digestive system. Moreover, these microorganisms may aid in the digestion of sugars and proteins, which is especially noteworthy given that infants have a shorter gastrointestinal transit time than adults [26,27].

### 1.2. Origin of Breast Milk Bacteria

The source of bacteria found in breast milk is not yet fully understood, but their existence is associated with a perinatal period that begins in the third trimester of pregnancy and lasts through lactation. Various studies have suggested the potential origins of the human milk microbiota, including colonization from the mother’s skin, the infant’s mouth during breastfeeding, or through the entero–mammary pathway from the mother’s gut. Additionally, it has been reported that a commensal microbiota exists in human breast tissue, indicating that certain microorganisms reside in the breast tissue and may potentially colonize the milk ducts [28,29].

The composition of milk microbiota can be influenced by various factors, including the duration of lactation, the mother’s dietary habits and nutritional status, the method of delivery, the gestational age of the baby, the geographical location, and the usage of antibiotics or other medications [28].

### 1.3. Human Milk Oligosaccharides (HMOs)

Human milk oligosaccharides (HMOs) are a group of diverse and unique glycans found abundantly in human milk. Initially identified as a prebiotic “bifidus factor”, HMOs act as a metabolic fuel for beneficial bacteria, contributing to a healthy intestinal microbiota in breastfed infants. Moreover, HMOs can influence the responses of epithelial and immune cells, leading to the reduced excessive infiltration and activation of immune cells in the mucosal lining. Additionally, HMOs offer the infant sialic acid, a potentially crucial nutrient for brain development and cognitive functions. The amount and composition of these oligosaccharides vary among women and throughout the lactation period [30]. Colostrum is a yellowish liquid produced by the mammary gland a few days before and after giving birth. It contains approximately 20–25 g/L of HMO [31,32]. As milk production progresses, the HMO levels decrease to about 5–20 g/L [30,31,33,34,35,36], which remains higher than the concentration of the total milk protein. Mothers who give birth prematurely have even higher levels of HMO in their milk compared to mothers who give birth at full term. 

Human milk oligosaccharides (HMOs) consist of five different monosaccharides: glucose (Glc), galactose (Gal), N-acetylglucosamine (GlcNAc), fucose (Fuc), and sialic acid (Sia). The primary or most prevalent form of sialic acid in HMOs is N-acetylneuraminic acid (Neu5Ac). The process of HMO biosynthesis seems to follow a fundamental pattern: all HMOs share a common core structure, starting with lactose (Galβ1-4Glc) at their reducing end. This core structure can be extended by the addition of either β1-3- or β1-6-linked lac-to-N-biose (Galβ1-3GlcNAc-, known as the type 1 chain) or N-acetyllactosamine (Galβ1-4GlcNAc-, known as the type 2 chain) [37].

### 1.4. Glycoside Hydrolases in Lactic Acid Bacteria

Carbohydrates are the most abundant biomolecules on Earth. They play a huge diversity of roles in living organisms, ranging from structural elements in cell walls (e.g., cellulose, chitin, starch, or glycogen) to cell–cell recognition processes of non-photosynthetic cells [38]. The vast amount and diversity of carbohydrate-based structures in nature require a large group of enzymes responsible for their synthesis, degradation, and modification [39]. These specialized enzymes are referred to as carbohydrate-active enzymes (CAZymes), including glycoside hydrolases (GHs), polysaccharide lyases (PLs), and glycosyltransferases (GTs). They make up approximately 1–2% of the genome of all living organisms. Oligosaccharides from human milk and other prebiotic sugars received through breast milk or formula remain unreconstructed in the gastrointestinal tract of the newborn. Hydrolysis of these sugars occurs primarily through enzymes secreted by the gastrointestinal microbiome.

### 1.5. HMOs Digestion

Milk oligosaccharides (OSs) have received significant attention for their anti-infective properties, anti-inflammatory activity, and regulatory activity of gut microbiome diversity and composition. Of these properties, the anti-infective potency of OSs is remarkable in protecting infants from various pathogenic bacteria [2].

However, it is essential to study the ability of different strains of bacteria to digest oligosaccharides from breast milk. Until now, two significant categories of beneficial microorganisms present in the gastrointestinal tract of infants, namely *Bifidobacterium* and *Bacteroides*, have been identified as significant recipients of breast milk glycans. The investigation of how *Bifidobacterium* obtains and metabolizes human milk oligosaccharides (HMOs) has gained substantial attention in recent times due to the high presence of this genus in the feces of breastfed infants. Numerous bifidobacterial genomes contain a considerable number of genes responsible for processing and transporting oligosaccharides, which are typically clustered within well-preserved regions.

The ability of certain *Bifidobacterium* strains to efficiently use HMOs suggests that the production of milk oligosaccharides by the mother may be a strategy to ensure the presence of this group of bacteria in the infant’s gut. The consumption of HMOs by *Bacteroides* species suggests that milk glycans may attract multiple groups of intestine-adapted mutualists to the infant’s intestine [1].

Our research aims to investigate the probiotic potential of strains isolated from breast milk and follow the enzymatic hydrolysis of breast milk oligosaccharides and other prebiotic sugars by hydrolyzing the enzymes of lactic acid bacteria isolated from breast milk. For this purpose, we isolated different primarily potential strains as well as oligosaccharides from human milk and compared the absorption of oligosaccharides with other prebiotic sugars by the specific strains.

## 2. Materials and Methods

All subjects gave their informed consent for inclusion before they participated in the study. The study was conducted following the Declaration of Helsinki, and the protocol was approved by the Ethics Committee of the University of Plovdiv (No/06.10.2022).

### 2.1. Subjects and Sample Collection

Samples were taken from 30 mothers who were breastfeeding their infants during the first to sixth month after giving birth. The participants were healthy lactating women who had given birth to full-term babies either through vaginal delivery or Caesarean section (C section). The inclusion criteria for the study required the gestation age to be at least 37 weeks and the birth weight to be greater than 1500 g.

To collect the milk samples, the mothers were instructed to clean their breasts with water and then express 15–20 mL of milk into a sterile container provided to them. The collected milk was stored at 4 °C until it was delivered to the laboratory. The processing of the milk samples took place within 24 h of donation.

### 2.2. Human Milk Oligosaccharides, Prebiotic Oligosaccharides, and Polysaccharides

In this study, we used human milk oligosaccharides (HMOs) provided by Friesland Campina (2-fucosyllactose) and oligosaccharides that we isolated, as well as the following oligosaccharides and polysaccharides: lactulose (Calbiochem, San Diego, CA, USA), galactooligosaccharide (GalOS) (Yakult, Tokyo, Japan), glucose, galactose and mucin (Sigma-Aldrich, St. Louis, MO, USA). 

### 2.3. Bacterial Isolation and Identification

The bacterial samples were collected and identified using conventional laboratory techniques, which involved the standard procedure for isolating bacteria from body fluids or anaerobic cultures. For anaerobic bacteria, the isolates were obtained by placing the samples on *Lactobacillus* de Man Rogosa and Sharpe (MRS) agar plates. Subsequently, these plates were kept in anaerobic conditions at 37 °C for 72 h [40].

Microbial DNA was extracted from 56 different bacterial strains from 30 breast milk samples using a modified protocol from the DNeasy Powerfood Microbial kit (Qiagen, UK). DNA templates were amplified by PCR using the universal primers amplifying a 1000-bp region of the 16S rRNA gene: 616 V: 5′-AGAGTTTGATYMTGGCTCAG-3′ and 699R: 5′-RGGGTTGCGCTCGTT-3′. The 616 V and 699R primers, Taq DNA polymerase, and dNTP mix were obtained from Thermo Fisher Scientific (Waltham, MA, USA). Amplicons were purified using the commercial Metabion GmbH mi-PCR Purification Kit (Planegg, Germany), followed by sequencing reactions using the BigDye Terminator v3.1 Cycle Sequencing Kit (Applied Biosystems, Waltham, MA, USA), with a premixed format. The resulting sequences were automatically aligned, visually inspected, and compared with the online tool BLAST (http://blast.ncbi.nlm.nih.gov/Blast.cgi, 5 June 2023). The strain was identified based on the highest scores.

### 2.4. Enzyme Analysis

The enzymatic activities of the strains were determined after isolation by centrifugation and lysis with 1 mL disintegrating buffer (containing 0.05 M sodium acetate buffer pH 7.5, 0.03 M NaCl, and 2% glycerol) and a UP 50 H Ultrasonic Processor (Hielscher, Ultrasound Technology). Then, the samples were centrifuged, and the supernatants were used to measure the activity of α-galactosidase, β-galactosidase, α-glucosidase, and α-fucosidase. 

The α-galactosidase activity was determined by the method of Petek et al. (1969) [41] with some modification, as the amount of p-nitrophenol (pNP) released by the degradation of pNP-α-D-galactopyranoside substrate (Sigma-Aldrich). The reaction mixture was composed of 250 μL 5.5 mM pNP-α-D-galactopyranoside substrate in 50 mM KH_2_PO_4_ buffer (pH 6.8) and 100 μL of the bacterial lysate. The total volume was brought up to 450 μL with distillate water and incubated for 20 min at 37 °C. The reaction was stopped by adding 2 mL of 1 M Na_2_CO_3_. The amount of the released pNP was measured spectrophotometrically at 405 nm.

The β-galactosidase activity was determined by the method of Lim and Chae (1989) [42] with some modification, as the amount of o-nitrophenol (oNP) released by the degradation of the oNP-β-D-galactopyranoside substrate (Sigma-Aldrich). The reaction mixture was composed of 250 μL 5.5 mM oNP-β-D-galactopyranoside substrate in 50 mM KH_2_PO_4_ buffer (pH 6.8), 100 μL of the bacterial lysate, and 100 μL distillate water. The mixture was incubated for 20 min at 37 °C. The reaction was stopped by adding 2 mL of 1 M Na_2_CO_3_. The amount of the released oNP was measured spectrophotometrically at 405 nm.

The α-glucosidase activity was assayed by the method of Dewi et al. (2007) [43] with some modification as the amount of pNP released by the degradation of the pNP-α-D-glucopyranoside substrate (Sigma-Aldrich). The reaction mixture was composed of 250μL 5.5 mM pNP-α-D-glucopyranoside substrate in 50 mM KH_2_PO_4_ buffer (pH 6.8) and 100 μL of the bacterial lysate. The total volume was brought up to 450 μL with distillate water and incubated for 20 min at 37 °C. The reaction was stopped by the addition of 2 mL of 1 M Na_2_CO_3_. The amount of the released pNP was measured spectrophotometrically at 405 nm.

The enzyme reaction (α-fucosidase activity) was carried out under the following conditions: 100 mM citrate buffer, pH 6.5, containing a 10 mM solution of p-nitrophenyl α-L-fucopyranoside (PNP-FUC) (prepared by diluting p-nitrophenyl α-L-fucopyranoside, Sigma cat. no. N3628) with distilled water and the appropriately diluted cells at 37 °C. The reaction was stopped with 200 mM borate buffer, pH 9.8. One unit of activity converted 1.0 µmole of p-nitrophenyl α-L-fucopyranoside to p-nitrophenyl and L-fucopyranoside in one minute at pH 4.0 and 37 °C. The p-nitrophenyl content was determined spectrophotometrically at a 405 nm wavelength on a Beckman Coulter DU 800 spectrophotometer (Beckman Coulter, Inc., Brea, CA, USA).

All experiments for the determination of enzyme activity were performed in triplicate.

### 2.5. Protein Content

The protein content in all the samples was determined by the method of Bradford (1976) [44] using bovine serum albumin (Sigma-Aldrich) as a standard. Spectrophotometric analyzes were performed on a Beckman Coulter DU 800 spectrophotometer (Beckman Coulter, Inc., Brea, CA, USA), and all experiments were performed in triplicate.

### 2.6. Human Milk Oligosaccharides Isolation

The breast milk was first thawed at room temperature and then thoroughly mixed. Next, it was centrifuged at 6000× *g* for 45 min at 4 °C. After this step, the solid fat from the top was separated and combined with one volume of distilled water at 37 °C. The mixture of lipid and water was gently swirled for several minutes and then subjected to another round of centrifugation at 4 °C. This process was repeated a second time, and all three aqueous phases were combined before proceeding to the next step.

To the defatted milk, two volumes of ethanol were added, and the mixture was left to stand overnight at 4 °C. As a result of this process, the protein precipitated, which was removed by centrifugation at 5000× *g* for 30 min at 4 °C. The sediment from the centrifuge tubes was then dissolved in two volumes of a mixture containing 2 parts ethanol and 1 part water. The solution was gently swirled to facilitate mixing and then subjected to another round of centrifugation at 5000× *g* for 30 min at 4 °C. This step was repeated a third time. The supernatants from all three precipitation/centrifugation rounds were combined, and the ethanol was removed from the mixture using rotary evaporation.

To digest the lactose, the milk extract was buffered by the addition of 40% *v*/*v* 0.123 M phosphate buffer, pH 6.5, an appropriate amount of β-galactosidase (Sigma Aldrich) was added based on the estimated lactose concentration and quoted enzyme activity, and the sample was incubated for 3 h at 37 °C. After lactose digestion, the particulates were removed via filtration with a vacuum.

The monosaccharides and any remaining disaccharides were extracted using solid-phase extraction, employing graphitized carbon columns. The process involved loading the columns with two volumes (120 mL) of the sample, followed by rinsing with two volumes of deionized water. The elution was performed using two volumes of a solution containing 60% acetonitrile and 40% water, with 0.01% trifluoroacetic acid. Finally, the columns were rinsed again with two volumes of deionized water. Refer to Figure 1 for a visual representation of the procedure [45].

## 3. Results

The primary screening was performed on MRS at a temperature of 37 °C, during which there were 124 isolates of putative lactic acid bacteria. Under the microscope, 36 isolates were characterized as rod-shaped, and the remaining 88 were cocci. All isolates were Gram-positive. For catalase activity, all the strains were negative (no bubbles were observed), (Figure 2).

### 3.1. Biochemical Characterization of the Obtained Isolates

#### Metabolism of Different Carbohydrate Sources

All 124 isolates were cultured on modified media with different carbohydrate sources to determine their ability to metabolize them. Separately, strains with mucosal growth were screened when cultured on Dols medium containing sucrose. All the isolates grew well on the MRS medium and could assimilate glucose. Approximately 75% of the breast milk isolates digested lactose at 2% and 5% concentrations. About 50% of the isolates digested sucrose at 2% and 5% concentrations, (Figure 3). 

None of the strains showed growth on only one medium, indicating that the isolated strains can assimilate different carbohydrate sources depending on the environment.

For further identification by 16S RNA and sequencing, 28 isolates from breast milk were selected that showed growth on MRS medium and media with 2% and 5% lactose at an OD above one at 24 h of culture and acidified to a pH lower than 4.5.

Figure 4a presents a hierarchical clustering of the selected 28 strains using optical density (OD) values after growth on different nutrient media. The OD600 provides information on the development of microorganisms after 24 h of cultivation. Almost all strains grew the fastest and had the highest optical density values after cultivating MRS with 2% glucose. Most of the selected 28 strains grew better on a lactose-containing medium than on sucrose. Perhaps it is no coincidence that there are strains in breast milk that absorb lactose well.

After colonization in the newborn’s intestinal tract, they would support the absorption of the high amounts of lactose arriving with the breast milk.

The selected 28 strains were identified after performing sequencing and Blast analysis. Then, it was found that 60% of them were representatives of the species *Lactobacillus fermentum*, 14% of *Enterococcus faecalis*, 18% of *Enterococcus faecium*, 4% of *Enterococcus lactis,* and 4% *of Lactobacillus lactis* (Figure 4b). Human milk contains several bacterial components, and *Bifidobacterium* and *Lactobacillus* species (spp.) are recognized as playing significant potential probiotic roles [12]. 

We next examined the ability of the isolated strains to digest the specific oligosaccharides and other prebiotics present in breast milk, including the ability to digest the glycans that are part of the mucin.

### 3.2. Enzyme Profile of the Selected Strains

#### 3.2.1. Effect of 2% Lactose and 8% Lactose on the Activity of β-Galactosidase

We monitored the dynamics of the β-galactosidase activity values in 12 studied strains. In Figure 5, the values of the cell-bound β-galactosidase activity are presented. The enzyme activity values after cell lysis were significantly lower. The *Lactobacillus* strains showed higher activities compared to the *Enterococcus* strains. The most increased activity was reported in the strain *Lactobacillus fermentum* st 8 on 8% lactose. This is proof that, as potential probiotics, the representatives of the genus *Lactobacillus*, through their β-galactosidase activity, actively participate in the breakdown of lactose and thus support absorption by the newborn.

Lactose represents approximately 8% of all sugars contained in breast milk, and its value may vary depending on the health of the mother, the period of lactation, and other physiological characteristics. Functional gastrointestinal disorders in newborns are common and can be caused by various reasons. A large problem is the so-called lactose intolerance. In this case, modeling intestinal flora supporting lactose absorption is particularly important. In this regard, establishing the degree of lactose absorption by the studied strains and related enzyme activities is of particular importance. Beta-galactosidase, also referred to as lactase, is an enzyme responsible for breaking down lactose through hydrolysis. When there is an excessive amount of lactose in the intestine, it can result in the dehydration of tissues and hinder the absorption of calcium due to the reduced acidity, leading to symptoms such as diarrhea, flatulence, and cramps [47].

Several experts propose enhancing the gut microbiota by introducing *Bifidobacterium longum* BB536 and *Lactobacillus rhamnosus* HN001, along with vitamin B6, for individuals with lactose deficiency. They emphasize the significance of specific probiotics and vitamin B6 in easing symptoms and addressing gut imbalances in lactose-intolerant patients experiencing persistent functional gastrointestinal issues [48].

#### 3.2.2. Metabolism of Galactooligosaccharides (GalOs), Lactulose, and Human Milk Oligosaccharides (HMOs) by the Studied Strains

The primary objective of this research was to examine the capacity of LAB (lactic acid bacteria) to ferment specific types of HMOs (human milk oligosaccharides), components of HMOs, GOSs (galactooligosaccharides), and lactulose. The study focused particularly on understanding the significance of β-galactosidases in the digestion of these oligosaccharides. All the chosen strains of LAB were isolated from breast milk. They were found to be part of the normal intestinal flora in newborns (*L. fermetum* st5, *L. fermetum* st6, *L. fermetum* st8, *L. fermetum* st10, *L. fermetum* st22, *L. fermetum* st24, *L. fermetum* ss5, and *L. fermetum* ss6). The selected strains showed good cell growth on the medium with 2% lactose.

We followed the β-galactosidase, α-glucosidase activity, and β-glucosidase activity of the selected only *Lactobacillus* strains. The primary trend that can be reported is that there was a strain-dependent possibility in the assimilation of the studied sugars, as shown in Figure 6.

The highest specific β-galactosidase activity was reported for strain *L. fermentum* St5 (2.4 U/mg), when cultured on a medium containing HMOs. Thus, HMOs were the strongest inducers for the transcription of the genes encoding β-galactosidase and the subsequent translation and secretion of the enzyme in active form.

The alpha-glucosidase activity was reported only when some strains were cultivated on a medium containing 2% galactooligosaccharides and on a medium with 2% lactulose. Such activity was not reported in any of the tested strains after cultivation in a medium containing 2% human milk oligosaccharides (HMOs), as shown in Figure 6.

Nondigestible carbohydrates present in the human diet can reach the large intestine without being broken down or absorbed. In this region, they serve as a source of energy for gut bacteria and also bring about changes in the composition of the gut microbiota. These carbohydrates fall into two categories: nonstarch polysaccharides, commonly known as “dietary fiber”, and certain oligosaccharides, such as lactulose, fructooligosaccharides, and galactooligosaccharides (GOS), which are considered prebiotics [49]. As per Gibson et al.’s findings, prebiotics are functional oligosaccharides that have the ability to selectively stimulate the growth of beneficial bacteria in the host, particularly *Bifidobacterium* and *Lactobacillus* [50,51,52,53,54,55]. Both dietary lactulose and fructooligosaccharides have been shown to protect the host from enteric pathogens and reduce the risk of colon cancer [56].

Bacteria can grow on oligosaccharides only when the latter are degraded into monosaccharides and metabolized for energy production. Therefore, transporters and glycoside hydrolases are vital factors in utilizing oligosaccharides [57,58].

Lactulose, recognized as the “bifidus factor” since the 1950s, is a straightforward disaccharide made up of galactose and fructose linked together through a β-1,4-glycosidic bond [59].

Bifidobacteria dominate the feces of healthy breast milk- or formula-fed infants [60,61]. *Bifidobacterium longum* subsp. infantis, whose genome possesses several clusters predicted to act on HMOs, is especially well adapted to metabolize HMOs [20]. Other bifidobacteria can ferment HMOs and components of HMOs to various extents [19,62]. Human milk is rich in lactose, and the presence of high lactose levels in human milk leads to the activation of lacZ in *Bifidobacterium longum* during its growth. This activation suggests that β-galactosidases play a role in breaking down lactose and human milk oligosaccharides (HMOs), releasing terminal nonreducing galactose units [63].

When compared to *Bifidobacterium infantis*, *Lactobacillus gasseri* shows limited capability in digesting HMOs.

It was interesting to follow the dynamics of the enzyme activities in the selected strains using the so-called resting cells from them. After cultivation and subsequent centrifugation, the cells were separated. Two percent solutions of lactulose, galactooligosaccharides, and oligosaccharides isolated from breast milk were added to the cells of the individual strains. The hydrolysis of the indicated sugars under the action of the enzyme β-galactosidase was followed. The results are presented in Figure 7.

In the presence of lactulose, however, the β-galactosidase activity and differences in activity varied greatly among the strains. The highest β-galactosidase activity was reported in the strain *L. fermentum* St22 in the presence of 2% lactulose at the third hour from the start of the process.

In the presence of 2% HMOs, the highest values of the β-galactosidase activity were recorded at 3 h from the start of hydrolysis, and a gradual decrease in activity was reported. The enzyme activity at each individual hour from the beginning of hydrolysis was compared. A general trend observed for the three substrates used was that the highest β-galactosidase activity was recorded at 3 h from the incubation of the resting cells and gradually decreased until 9 h. This proves, on the one hand, the inducible nature of the secreted β-galactosidase in the studied lactobacilli and, on the other, the different degrees of induction in the presence of the various substrates. A likely cause is either the substrate depletion or the product inhibition of the enzyme activity.

#### 3.2.3. Beta–Galactosidase Activity in the Presence of 2% Mucin, 2% HMOs, and 2% 2-Fucosyllactose 

We followed the beta-galactosidase activity after culturing the strains on media containing 2% mucin, 2% human milk oligosaccharides, and 2% 2-fucosyllactose.

For the study and as a carbon source for one of the cultivations, we used a product 2-fucosyllactose 2’-FL (2’-FL) provided by Friesland, The Netherlands.

The product had the following content after analysis in a certified laboratory: 2’-fucosyllactose-93%, lactose-0.5%, allolactose-1%, glucose < 0.1%, galactose < 0.1%, fucose 0.2%, protein < 0.01%, nitrite 0.1 mg/kg, and nitrates 0.9 mg/kg.

Mucin is the main mucus component, forming a protective barrier between the microbiota and immune cells. Mucin-type O-glycans alter the diversity of gastrointestinal microorganisms.

Mucin, particularly the intestinal mucin MUC2 O-glycans, likely plays a significant role in determining the colonic flora of a particular species. Moreover, certain structures present on mucin-type O-glycans, which interact with bacterial agglutinin-like adhesins, likely affect the makeup of intestinal colonies. The process of bacterial adhesion is closely linked to the biological function of intestinal mucus. O-glycans serve as an initial point of attachment for bacteria, including specific pathogens. Abundant glycosylating enzymes, such as glycosidases, typically characterize the commensal bacteria from the gut microbiome [64]. When dietary glycans are scarce, mucolytic activity bacteria can degrade and metabolize mucin-type O-glycans. Therefore, we analyzed the ability of the lactobacilli strains isolated and studied by us to digest glycans from mucin using glycolytic enzymes. We used mucin at a concentration of 2%, which we put into the culture medium. Then we analyzed the activity of the β-galactosidase enzyme and compared it with the growth activity of the same strains in media with other carbon sources, as shown in Figure 8.

The human milk oligosaccharides (HMOs) we used were isolated from breast milk according to the methodology indicated in the Section 2. Human milk oligosaccharides, followed by 2-fucosyllactose are the best inducers of beta-galactosidase.

All strains germinated on a medium with a single carbohydrate source of 2% mucin. This is evidence that the strains we isolated from breast milk samples can digest specific glycans from the mucin structure.

#### 3.2.4. Alpha-Fucosidase Activity in the Presence of 2% Mucin, 2% HMOs, and 2% 2-Fucosyllactose

Degradation of various probiotic oligosaccharides, including mucin-bound glycoproteins, as well as oligosaccharides present in breast milk and having a significant impact on the colonization of different bacterial species in the digestive tracts of newborns, is carried out by a unique set of enzymes (β-galactosidase, fucosidase, β-hexosaminidase, lacto-N-phosphorylase, and sialidase) [55].

Alpha-fucosidase catalyze the release of α-linked fucose residues. Most fucosidases harbored by Bifidobacteria belong to CAZY families 29 or 95 (GH29 and GH95). The first bifidobacterial cloned and characterized fucosidase was AfcA from *B. bifidum*, an extracellular cell-anchored α-1,2-specific fucosidase [65].

Regarding this matter, we investigated the fucosidase activity of certain strains, as depicted in Figure 9. We cultured the strains on media containing human milk oligosaccharides, mucin, and fucosyllactose.

We found the highest α-fucosidase activity by *L. fermentum* ss 8 strain (2.5 U/mg) in the presence of HMOs in media.

In contrast to Bifidobacteria, the utilization of human milk oligosaccharides (HMOs) by members of the *Lactobacillus* genus is a relatively new area of research, and their ability to utilize this carbon source seems somewhat limited. Intestinal lactobacilli typically possess a significant number of glycosylhydrolases in their genomes, which are likely responsible for processing carbohydrates obtained from the diet. While many characterized sugar transporters are involved in transporting various types of mono and disaccharides, not all of them are necessarily linked to host glycans, with the exception of GlcNAc. The capacity to break down terminal sugars from HMOs is also diminished in lactobacilli.

Up until now, researchers have identified and studied three α-L-fucosidases (AlfA, AlfB, and AlfC) belonging to the GH29 family in lactobacilli. These enzymes are mainly found in strains of the closely related *L. casei–paracasei–rhamnosus* group. Different strains within this group may carry one, two, or all three of these enzymes. All three enzymes are homotetramers [66] located inside the cell, and they rely on the transportation of fucosylated carbohydrates to carry out their functions.

## 4. Discussion

The distinct composition of microorganisms, such as *Lactobacillus* and *Bifidobacterium,* within the mother’s milk, is crucial in shaping the newborn’s microbiome and establishing protective defenses against harmful pathogens. We were able to isolate from the different breast milk samples we collected different strains of microorganisms mainly from *Lactobacillus fermentum* and *Lactobacillus brevis*. We also optimized the scheme for the isolation of oligosaccharides from breast milk by putting the lyophilized product into different media for cultivation. They referred mainly to the heterofermentative lactobacilli and showed specificity regarding the secretion of enzymes promoting the absorption of HMOs. It was important for us to establish the presence of the different glycolytic enzymes with which the strains we isolated could degrade specific prebiotic sugars.

We focused primarily on the activity of the enzyme beta-galactosidase, as previous studies have identified this enzyme as one of the five major enzymes involved in the absorption of milk oligosaccharides. We found that the presence in the medium of human milk oligosaccharides from breast milk was the strongest inducer for the synthesis of this enzyme. The highest specific β-galactosidase activity was reported for strain *L. fermentum* St5 (2.4 U/mg).

Also, we found that all the lactobacilli strains we studied in detail were able to digest mucin-linked glycans. The degradation of these sugars is perhaps a built-in defense mechanism in cases where other sugars are lacking in the environment.

We were also able to examine the fucosidase activity in some of the isolated strains. We recorded the highest values (2.5 U/mg in *L. fermentum* ss8), when oligosaccharides isolated from breast milk were present in the medium. It is clear from our study that the induction of the two important enzymes for the assimilation of oligosaccharides from breast milk, namely fucosidase and beta galactosidase, was highest when these sugars were present in the medium.

It is important to further investigate this ability in pathogenic strains of microorganisms as well as to study in depth the other four enzymes mediating the absorption of oligosaccharides from breast milk.

*Lactobacilli* and *Bifidobacteria* supplied with breast milk are the first colonizers in most cases in the gastrointestinal tract of the newborn. The presence and study of different genes for the synthesis of different enzyme systems and transporters of different sugars in this type of bacteria is essential.

This will lead to a better understanding of the colonization mechanisms of potential probiotics delivered through breast milk to the infant.

## Figures and Tables

**Figure 1 microorganisms-11-01904-f001:**
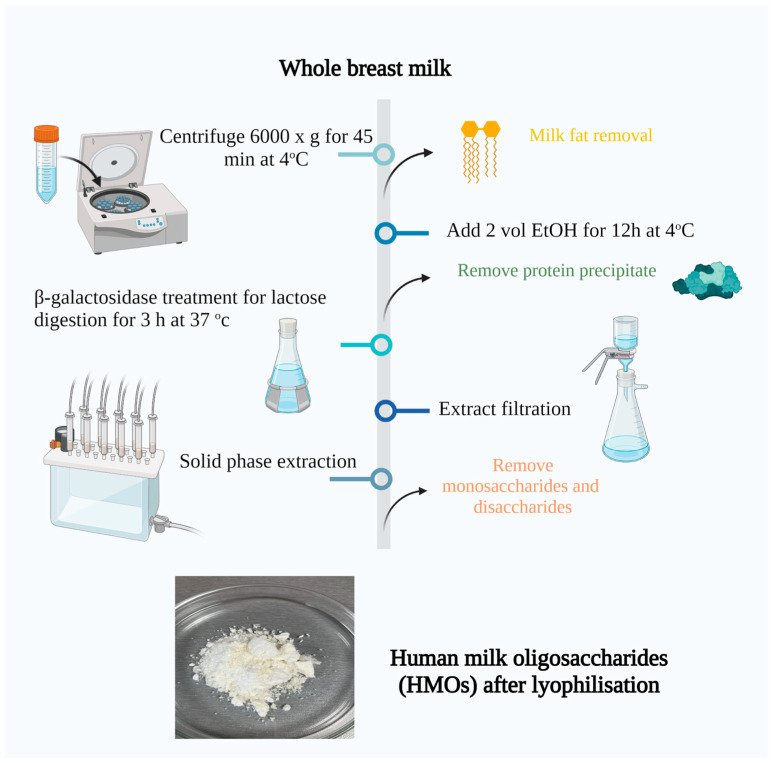
Isolation of human milk oligosaccharides from breast milk. Created with “Biorender.com (accessed on 5 June 2023)”.

**Figure 2 microorganisms-11-01904-f002:**
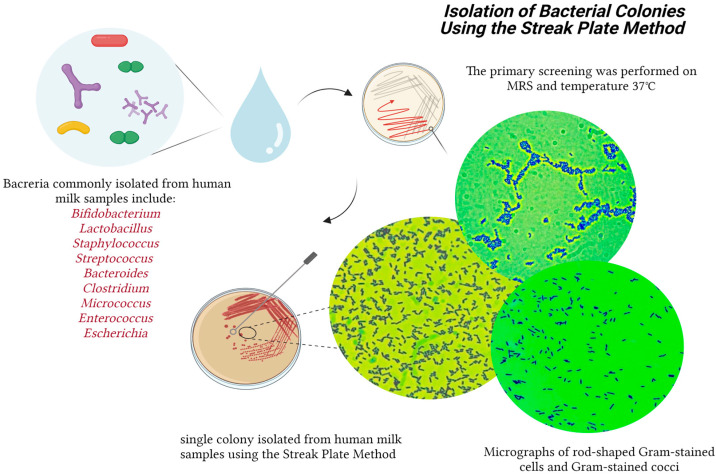
Isolation of the bacterial colonies from breast milk. Created with “Biorender.com (accessed on 5 June 2023)”.

**Figure 3 microorganisms-11-01904-f003:**
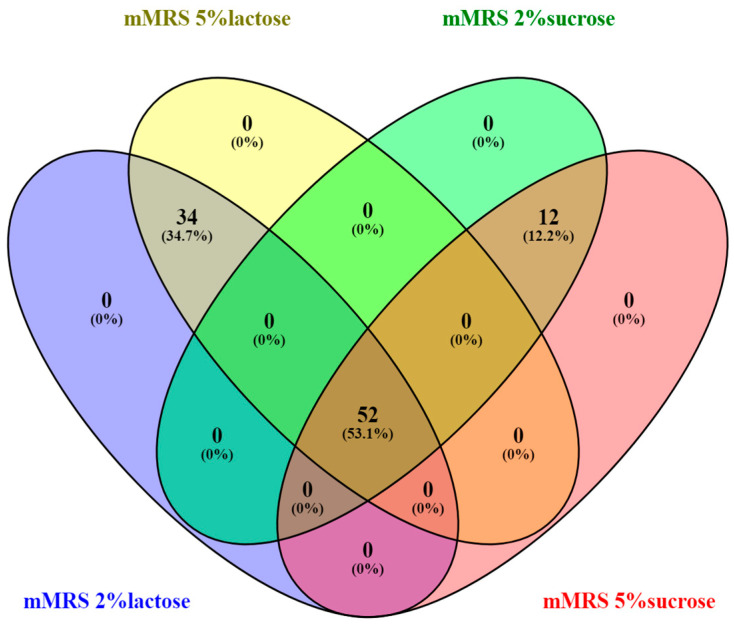
Venn diagram illustrating the overlap of 124 bacterial isolates based on their growth on different nutrient media Created by Venny2.1 Venny diagram Oliveros, J.C. (2007–2015) Venny. An interactive tool for comparing lists with Venn diagrams. “https://bioinfogp.cnb.csic.es/tools/venny/index.html (accessed on 5 June 2023)” [46].

**Figure 4 microorganisms-11-01904-f004:**
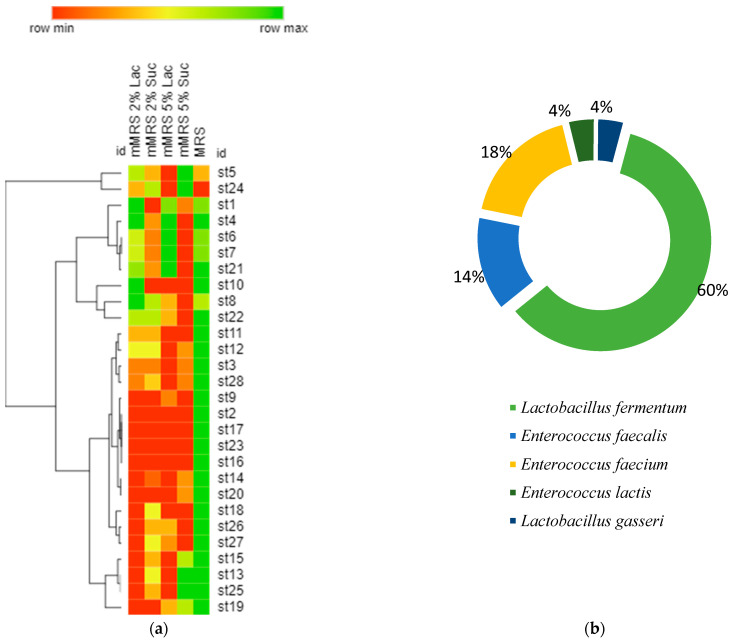
(**a**) Hierarchical clustering of the selected 28 strains using optical density (OD600) values after 24 h of growth on different nutrient media. (Row min represents value = 1, and row max represent represents the maximum reported value of OD = 1.8); (**b**) the percentage distribution of species after identification by sequencing and BLAST.

**Figure 5 microorganisms-11-01904-f005:**
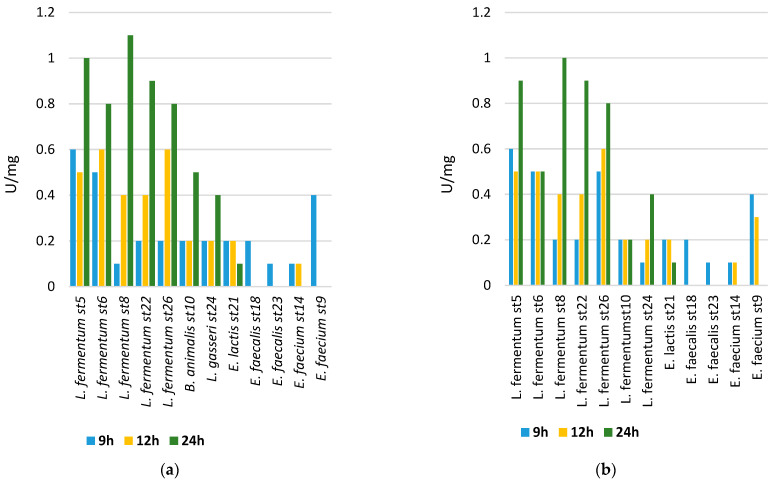
(**a**) The beta-galactosidase activity of cells from the cultured strains after cultivation on medium containing 8% lactose; (**b**) and 2% lactose.

**Figure 6 microorganisms-11-01904-f006:**
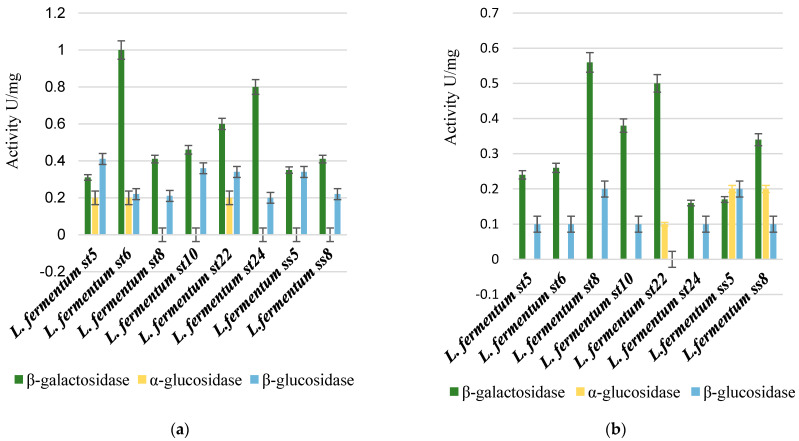
Comparison between β-galactosidase, β-glucosidase, and α-glucosidase enzyme activity after cultivation on 2% galactooligosaccharides (**a**), 2% lactulose (**b**), and 2% HMOs (**c**).

**Figure 7 microorganisms-11-01904-f007:**
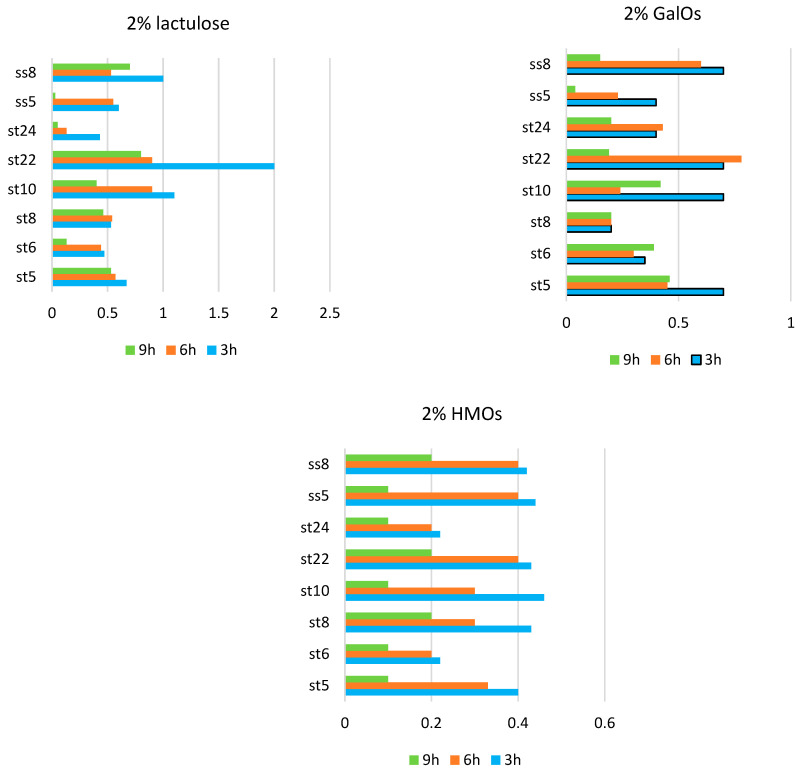
Beta-galactosidase activity in the resting cells in strains of *Lactobacillus fermentum* (st5, st6, st8, st10, st22, st24, ss5, and ss8).

**Figure 8 microorganisms-11-01904-f008:**
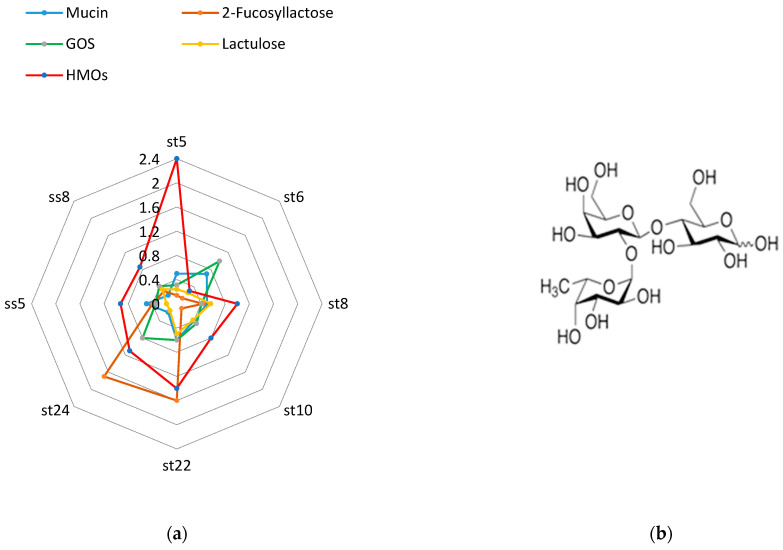
Beta-galactosidase activity after cultivation on mucin, 2-fucosyllactose, GOS, lactulose, and HMOs (**a**) and the structure of 2-FL (**b**).

**Figure 9 microorganisms-11-01904-f009:**
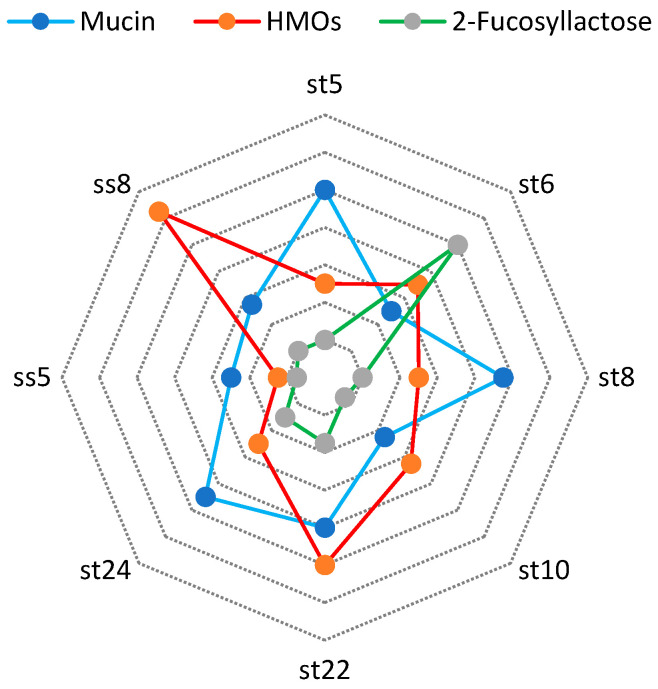
Alpha-fucosidase activity after cultivation on mucin, 2-fucosyllactose, and HMOs.

## Data Availability

The corresponding author can provide the supporting data for the findings mentioned in this study upon reasonable request.

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
