# Peer review of "The Enzymatic Hydrolysis of Human Milk Oligosaccharides and Prebiotic Sugars from LAB Isolated from Breast Milk"

_microorganisms, 2023, doi:10.3390/microorganisms11081904_

Round 1

Reviewer 1 Report

This manuscript focused on the enzymatic hydrolysis of HMO and other prebiotics by human-milk-derived-LAB. Here are my comments and suggestions.

1. The activities of β-galactosidase, α-glucosidase, and β-glucosidase were attached to great importance and determined in various experiments of the study. However, these three enzymes commonly exist in various strains. Is there any inevitable relation between the three enzymes and HMO hydrolysis? Or other prebiotics? If not, the multiple determinations throughout the article seem unnecessary and not persuasive enough to readers.

2. One of the objectives of the article was to “investigate the enzymatic hydrolysis of breast milk oligosaccharides and other prebiotic sugars by hydrolyzing enzymes of lactic acid bacteria isolated from breast milk”. For this purpose, the prebiotics hydrolysis should be confirmed by determining HMO consumption, characteristic product, or other certain evidence after the isolation and identification of the strains. It has been mentioned in lines 355-356 that “this study aimed to investigate the ability of LAB to ferment defined HMOs, HMO components, GOSs, and lactulose”. Was the fermentation ability confirmed by experiments? Please provide the corresponding results.

4. The introduction (a mini-review) was too redundant and involved too many aspects, which make it difficult to show the value of the study. Please simplify the introduction and make it more concise.

5. Readability of the article remains to be increased, including modifying inaccurate expressions and cutting down on redundant paragraphs. Check the manuscript thoroughly and try to present the crucial point in each paragraph, such as Line 14: “lactobacilli”; Line 52: “1 × 105 and 1 × 107”; Line 208: “α-galactosidase”; Lines 338-340: unnecessary paragraphing.

Author Response

Thank you for the review you provided!

 Regarding the questions from the review, I will try to provide detailed answers point by point:

  1. The three enzymes examined are indeed found in different bacterial strains and are crucial for digesting various prebiotic sugars. However, beta-galactosidase is the main enzyme involved in the hydrolysis of oligosaccharides from breast milk. In Figure 6, we attempted to present a comparison of the three enzymes when cultured with different sugars, including oligosaccharides from breast milk. This allowed us to demonstrate the inducible nature of the enzymes, as well as the high activities specifically of beta-galactosidase in the presence of breast milk oligosaccharides.

  2. The fermentation ability was determined and shown in Figures 7, 8, and 9. The bacterial strains were able to grow in a medium with a specific sugar as the sole source. The figures represent the enzyme activities of both beta-galactosidase and alpha-fucosidase, both of which are among the five essential enzymes involved in the metabolism of breast milk oligosaccharides.

  3. Thank you for the recommendation! We simplified the introduction.

  4. Thank you! We tried to remove all unnecessary paragraphs, including those pointed out by you.

Reviewer 2 Report

Dear autors,

In the presented manuscript, studies were carried out on the microbiota of mother's breast milk, biochemical characteristics of the isolated strains regarding their glycosidase activities.

The originality of the article lies in the authors' studies of the relationship between the glycoside hydrolase activities of the isolated strains and their ability to colonize, and hence the evaluation of their potential probiotic properties.

A protocol for the isolation of oligosaccharides in human milk has been optimized, which also describes the high quality of the studies.

The presented hypothesis is defended with adequate and appropriate analyses. The article is distinguished by a high level of originality and scientific significance.

 The presented approach allows, by examining the enzymatic activities of the strains, to evaluate the potential probiotic properties of lactic acid bacteria.

I reccomend accetpance of manuscript in the present form.

Author Response

Thank you for the review you provided